# A Mathematical Model of Deformation under High Pressure Torsion Extrusion

**Roman Kulagin [1,*], Yan Beygelzimer [1,2], Yuri Estrin [3,4], Yulia Ivanisenko [1], Brigitte Baretzky [1] and Horst Hahn [1]**

[1] Institute of Nanotechnology, Karlsruhe Institute of Technology, 76344 Eggenstein-Leopoldshafen, Germany; yanbeygel@gmail.com (Y.B.); julia.ivanisenko@kit.edu (Y.I.); brigitte.baretzky@kit.edu (B.B.); horst.hahn@kit.edu (H.H.)

[2] Donetsk Institute for Physics and Engineering named after A.A. Galkin, National Academy of Sciences of Ukraine, 03680 Kyiv, Ukraine

[3] Department of Materials Science and Engineering, Monash University, Clayton 3800, Australia; yuri.estrin@monash.edu

[4] Department of Mechanical Engineering, The University of Western Australia, Crawley 6009, Australia

* Correspondence: roman.kulagin@kit.edu; Tel.: +49-721-6082-8127

**Abstract:** High pressure torsion extrusion (HPTE) is a promising new mechanism for severe plastic deformation of metals and alloys. It enables the manufacture of long products with a radial gradient ultrafine-grained structure and of composite materials with a helical inner architecture at the meso and the macro scale. HPTE is very promising as a technique enabling light weighting, especially with magnesium, aluminium and titanium alloys. For the first time, this article presents an analytical model of the HPTE process that makes it possible to investigate the role of the various process parameters and calculate the distribution of the equivalent strain over the entire sample length. To verify the model, its predictions were compared with the numerical simulations by employing the finite element software QForm. It was shown that potential negative effects associated with the slippage of a sample relative to the container walls can be suppressed through appropriate die design and an efficient use of the friction forces.

**Keywords:** light metals; processing; severe plastic deformation; high pressure torsion extrusion; finite element model; equivalent strain; mechanical properties

## 1. Introduction

Recent years have seen a growth in popularity of the concept of architectured materials, which makes potential breakthroughs in materials science a realistic possibility [1–4]. This has prompted innovative developments in various areas of materials engineering and design [4]. In particular, the emerged paradigm of architectured materials offered an invigorating stimulus to the area of severe plastic deformation (SPD) technologies [5,6]. Not only can these technologies produce a submicron scale grain structure that provides metals and alloys with exceptional mechanical performance, but they can also be used to form various inner architectures of a workpiece at a mesoscopic and macroscopic scale [7–11]. In this regard, the SPD methods involving plastic torsion, such as high pressure torsion (HPT) [12,13], incremental HPT [14], twist extrusion [15], shear extrusion [16], high pressure torsion extrusion (HPTE) [17,18], torsion extrusion [19], three roll planetary milling [20], spiral equal channel angular extrusion [21], tandem process of simple shear extrusion, and twist extrusion [22]. Along with producing a submicron scale structure in the processed materials, these methods make it possible to form a helical architecture of various inclusions or reinforcements introduced in a workpiece

beforehand [7–9,11,23]. These kinds of structures often occur in nature [24], which suggests that they may be very beneficial and promising in materials engineering as well. In particular, ultrafine-grained (UFG) materials with helical inner architecture offer themselves for applications of magnesium, aluminium, and titanium alloys in lightweight structures with high specific strength [9].

Multi-scale structures produced by SPD techniques are controlled by deformation processes. Therefore, for obtaining submicron structured architectured materials with desired characteristics, specific parameters of the SPD process need to be established. In some cases, the deformation of a sample can be effectively controlled through the appropriate choice of the tool geometry and/or the regime of the sample motion. The major problem is that the SPD processes mentioned above are not controllable in this way. Undesirable slippage of the sample at the contact surface with the container may cause non-steadiness of these processes, thus making them ill-controlled. For example, slippage during HPT causes a decrease in strain and its non-uniform distribution over the sample thickness [25,26]. A mathematical model [27] that accounts for slippage makes it possible to predict such effects and design the process accordingly.

A detrimental effect of slippage also occurs during HPTE. In essence, in this process a cylindrical sample is moved through two joined coaxial containers, one of which is stationary and the other is rotated about its axis [17]. The contact friction force in the second container produces rotation of the sample, which is opposed by the friction force in the first one. For sufficiently long portions of the sample in each of the two containers, these forces are large enough to produce a torque that is necessary for plastic deformation of the material. When these portions are not long enough, the sample slips in the tangential direction, the character of the process becomes non-steady and the equivalent von Mises strain at its initial and final stages is reduced.

HPTE combines the benefits of HPT with the ability to process long samples in a semi-continuous way—a quality the conventional HPT does not possess. This makes HPTE interesting for industrial scale applications. For better control of the HPTE process, a container design with special holding elements was suggested [18]. For sufficiently large dimensions, this design ensures full suppression of sample slippage. However, long holding elements increase the non-uniformity of deformation of the end parts of the sample, which is undesirable. Thus, one is faced with the problem of finding a reasonable compromise between the conflicting requirements of maximizing the friction forces and minimizing the proportion of the non-uniformly deformed portion of the sample length.

Another crucial aspect of design of the HPTE process is ensuring that simple shear deformation occurs in a thin layer of the sample [28–30]. To solve these problems, we propose a simple mathematical model that makes it possible to calculate the optimal parameters of the HPTE process. The model is presented in the subsequent sections.

## 2. The Model

The principal features of the HPTE process are schematically illustrated in Figure 1, which also defines the quantities used in the calculations of the process.

In Figure 1c, the plane where the two joined containers (not shown in the drawing) meet is denoted by $S$. A cylindrical part of height $H$ indicates a zone in which the holding elements are located. We assume that at any time the sample consists of two stiff blocks: block 1 sitting in the stationary upper container and block 2 residing in the lower container that is rotated with an angular velocity $\omega$. Owing to tangential slippage of the sample relative to the walls of the containers, the angular velocities $\omega_1$ and $\omega_2$ of the rotation of the blocks differ from those of the containers. The sample is translated downwards with a speed $\vartheta$, so that the length of the first block is decreased and that of the second block is increased. The lengths of the parts of blocks 1 and 2 that lie outside of the holding elements are denoted $L_1$ and $L_2$, respectively. Obviously, they are related through $L_1 + L_2 = L - H$, where $L$ is the total sample length. The kinematics of the HPTE thus defined imposes rotation of blocks 1 and 2 with angular velocities $\omega_1$ and $\omega_2$. These are the quantities we set out to calculate.

Friction between the sample and the container walls gives rise to a stress $\tau = mk$, where $k$ is the shear yield stress of the material and $m$ is the coefficient of plastic friction. The plastic friction is assumed since large pressures are required to obtain high quality submicron microstructures, in which case such a model is more consistent with experiment than the classical Coulomb friction law [31].

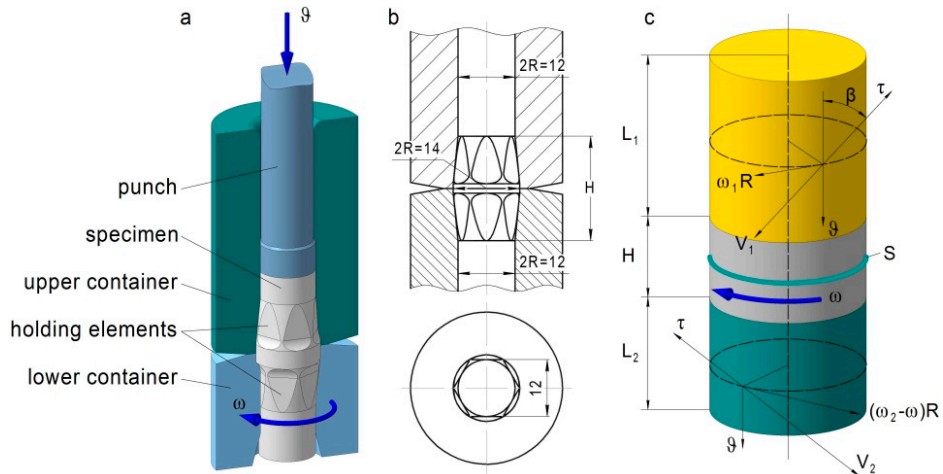

**Figure 1.** High pressure torsion extrusion: (**a**) schematics of the process, (**b**) design of containers and (**c**) definitions of the quantities for calculations. (unit: mm).

The material of the sample is assumed to be rigid-plastic and non-strain hardening. These assumptions are in keeping with the velocity field that experiences a discontinuity postulated below. We note that a strain hardening material does not allow for discontinuities. As was shown earlier (see, e.g., [31]), these assumptions yield meaningful estimates of the equivalent strain.

Expressions for the torques associated with the friction forces acting on blocks 1 and 2 can be obtained in the following way. According to solid mechanics, an elementary torque $dM_f$ acting on a surface area $dS$ of a sample is given by $dM_f = R \cdot \tau dS \cdot \sin\beta$, where $\beta$ is the angle between the direction of the friction force and the rotation axis and $R$ is the radius of the cylindrical channels of the containers. Considering that at any point on the sample surface the friction force is directed against the direction of movement of that point relative to the container surface and using $\sin\beta$ as determined from Figure 1, the friction torques $M_{f1}$ and $M_{f2}$ acting on blocks 1 and 2 are obtained upon integration of the elementary torques over the surfaces of the two blocks:

$$M_{f1} = 2\pi L_1 R^2 mk \frac{\omega_1 R}{\sqrt{\vartheta^2 + (\omega_1 R)^2}} \tag{1}$$

$$M_{f2} = 2\pi L_2 R^2 mk \frac{(\omega - \omega_2)R}{\sqrt{\vartheta^2 + (\omega - \omega_2)R)^2}} \tag{2}$$

The torques, produced by the upper and lower holding elements, $M_{h1}$ and $M_{h2}$, are assumed to be equal: $M_{h1} = M_{h2} = M_h$. Their magnitude $M_h$ is presented in the form $M_h = CM_T$, where

$$M_T = \frac{2}{3}\pi k R^3 \tag{3}$$

is the torque associated with the plastic twisting of the sample [31]; $C$ is a proportionality coefficient.

We further assume that for a given design of the holding elements, the magnitude of the torque they exert on the sample is proportional to their combined length, that is, that the relation $M_{h1} = (H/H_0)M_T$ holds, where $H_0$ is the smallest length of the holding elements with the same design, which produce the torque $M_T$.

The torques associated with friction forces at the end faces of the sample are neglected. Under this assumption the torque on the sample is underestimated, since in the real process the friction on its end surfaces produces torques of opposite sense in the upper and lower containers. As a result, the length of the deformed part of the sample is underestimated as well.

The magnitude of the resultant torques in the upper and lower containers is given by $M_{f1} + M_h$ and $M_{f2} + M_h$ respectively.

Let us consider two possible cases: (i) $\omega_1 = \omega_2 = \omega_0$ and (ii) $\omega_2 > \omega_1$.

In the first case, the sample moves as a whole, meaning that no plastic deformation occurs. In the second case, the tangential component of the velocity experiences a discontinuity at the plane where the two containers meet. Plasticity theory [31] tells us that this leads to simple shear in the material corresponding to a von Mises strain $e_M = \frac{[\vartheta_\tau]}{\sqrt{3}\vartheta_n}$. Here $[\vartheta_\tau]$ is the absolute value of the jump in the tangential component of the velocity and is the absolute value of the velocity component normal to the plane of discontinuity. In our case, the relations $[\vartheta_\tau] = r(\omega_2 - \omega_1)$ and $\vartheta_n = \vartheta$ hold. Substitution of these expression in the above formula for the von Mises equivalent strain yields

$$e_M = \frac{1}{\sqrt{3}} \cdot \frac{r(\omega_2 - \omega_1)}{\vartheta} \tag{4}$$

Variant (i) is realized when the torque applied to the sample is smaller than the torque associated with plastic torsion. In that case only the equality of the absolute values of the torques acting on the sample in the two containers, $M_{f1} + M_h = M_{f2} + M_h$, must hold, which can be re-written as

$$L_1 \frac{\omega_1}{\sqrt{\vartheta^2 + (\omega_1 R)^2}} = L_2 \frac{(\omega - \omega_2)}{\sqrt{\vartheta^2 + ((\omega - \omega_2)R)^2}} \tag{5}$$

We now use the equality $\omega_1 = \omega_2 = \omega_0$ in Equation (5) and obtain the following implicit relation for the normalized velocity $a = \vartheta/(\omega R)$ of the rotation of the entire sample:

$$\left(\frac{1}{\xi} - 1\right) \frac{\omega_0'}{\sqrt{a^2 + \omega \, r_0'^2}} = \frac{1 - \omega_0'}{\sqrt{a^2 + (1 - \omega_0')^2}} \tag{6}$$

where $\xi = L_2/(L - H)$ and $\omega_0' = \omega_0/\omega$.

Variant (ii) is realized if the condition

$$\begin{cases} M_{f1} + M_h = M_T \\ M_{f2} + M_h = M_T \end{cases} \tag{7}$$

is fulfilled. In this case the angular velocities of blocks 1 and 2 can be determined from a set of two equations: the torque equilibrium equation, Equation (6), and the relation expressed by Equation (7), which, after substitution of the expressions for the torques takes the form

$$2\pi L_2 R^2 mk \frac{(\omega - \omega^2)R}{\sqrt{\vartheta^2 + ((\omega - \omega^2)R)^2}} + M_h = \frac{2}{3}\pi k R^3 \tag{8}$$

Equation (8) can be re-written as

$$\xi m \frac{(1 - \omega_2')}{\sqrt{a^2 + (1 - \omega_2')^2}} = \frac{1}{3} \frac{R}{(L - H)} - \frac{M_h}{2\pi k R^2 (L - H)} = \frac{1}{3} \frac{R}{(L - H)}(1 - H') \tag{9}$$

where $\omega_2' = \omega_2/\omega$ and $H' = H/H_0$. The normalized frequency $\omega_2'$ can now be found from Equation (9). By combining Equations (4) and (9) the following equation for $\omega_1' = \omega_1/\omega$:

$$m(1 - \xi)\frac{\omega_1'}{\sqrt{a^2 + \omega_2'^2}} = \frac{1}{3}\frac{R}{(L - H)}(1 - H') \tag{10}$$

We now wish to determine the conditions for variant (ii) to apply. To that end, we analyze Equations (9) and (10).

A non-negative solution of Equation (10) reads

$$\omega_1' = \frac{a \cdot b}{\sqrt{(1 - \xi)^2 - b^2}} \tag{11}$$

where the parameter $b$ is defined by

$$b = \frac{R}{3m(L - H)}(1 - H') \tag{12}$$

It should be noted that Equation (9) is reduced to Equation (10) if the following substitutions

$$\omega_1' \rightarrow (1 - \omega_2') \text{ and } (1 - \xi) \rightarrow \xi \tag{13}$$

are made. As the condition $(1 - \omega_2') \geq 0$ is fulfilled, the solution of Equation (9) reads

$$\omega_2' = 1 - \frac{a \cdot b}{\sqrt{\xi^2 - b^2}} \tag{14}$$

From Equations (11) and (14) it follows that a solution exists only in the range of

$$b < \xi < 1 - b \tag{15}$$

As $\omega_2' > 0$ holds, Equation (14) puts a further restriction on the parameter $\xi$:

$$\frac{a \cdot b}{\sqrt{\xi^2 - b^2}} < 1 \tag{16}$$

This is tantamount to the inequality

$$\xi > b\sqrt{a^2 + 1} \tag{17}$$

Combining inequalities (15) and (17) we have

$$b\sqrt{a^2 + 1} < \xi < 1 - b \tag{18}$$

It can thus be concluded that for plastic deformation of a sample to occur, the inequality $b\sqrt{a^2 + 1} < 1 - b$ must be satisfied, i.e., the inequality

$$b < \frac{1}{1 + \sqrt{a^2 + 1}} \tag{19}$$

has to hold. Finally, substitution of the expressions defining the quantities $a$ and $b$ transforms inequality (19) to

$$\frac{L-H}{R} > \frac{1}{3m}(1-H')\left(1+\sqrt{\left(\frac{\vartheta}{\omega R}\right)^2+1}\right)$$

(20)

This inequality, expressed in terms of the container geometry and process parameters, is a basis for determining the conditions for plastic twisting of a sample under HPTE to occur.

## 3. Results and Discussion

We now analyse the model proposed for conditions close to real experiment [18]. The holding element design employed in [18] did ensure the occurrence of plastic torsion of samples. Let us set $H_0$ to be equal to the length of a holding element used in experiment ($H_0$ = 24 mm). The characteristic process parameters used in [18] and summarised in Table 1 will be adopted in the analysis to follow.

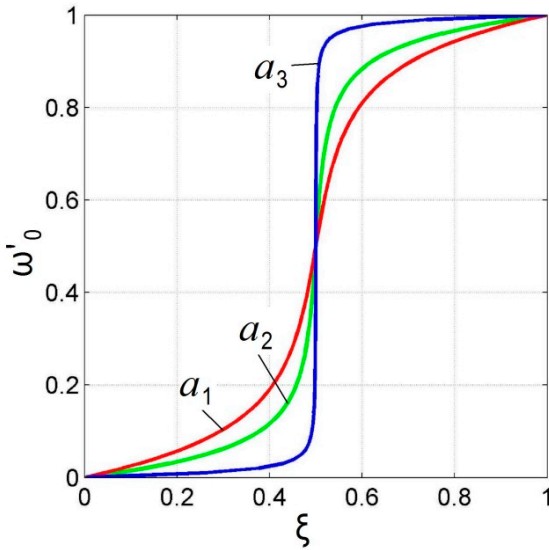

**Figure 2.** Dependence of the non-dimensional angular velocity of the sample, $\omega_0'$, on the position of the sample, $\xi$, in the container channel.

**Table 1.** High pressure torsion extrusion (HPTE) parameters for analysis of the model used to calculate the graphs presented in Figure 2.

| Regime | $\vartheta$, mm/min | $\omega$, rpm | $a$ |
|--------|--------|--------|--------|
| $a_1$ | 10 | 1 | 0.265 |
| $a_2$ | 5 | 1 | 0.133 |
| $a_3$ | 1 | 1 | 0.027 |

Figure 2 presents the dependence of $\omega_0'$ on $\xi$ obtained by solving Equation (6) for three different values of the parameter $a$.

As seen from the graphs, the angular velocity of a rigid sample translated through the joint plane between the containers varies from zero (when the entire sample sits in the stationary upper container) to $\omega$—the angular velocity of rotation of the lower container. The smaller the magnitude of the parameter $a$, the more precipitous is this transition from zero to $\omega$ at the point in time when the middle of the sample passes the joint plane between the two containers ($\xi$ = 0.5).

The dependence of the angular velocities obtained by solving Equations (9) and (10) on the process parameters is presented graphically in Figure 3, which depicts $\omega_1'$ and $\omega_2'$ vs. $\xi$.

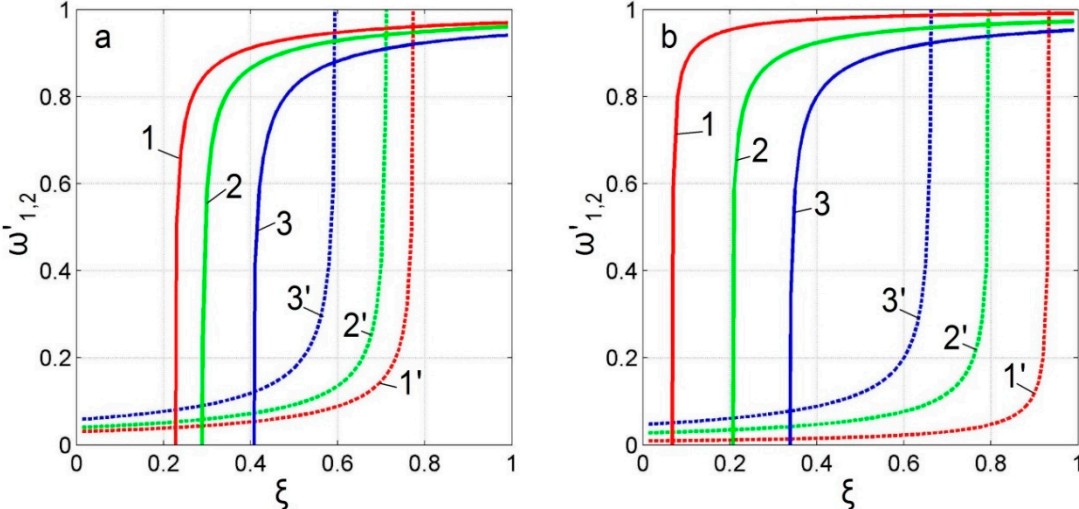

**Figure 3.** Dependence of the normalized angular velocities of the two parts of a sample (blocks 1 and 2), $\omega_1'$ (1, 2, 3) and $\omega_2'$ (1′, 2′, 3′) on the position of the sample in the HPTE container channel, $\xi$, for (**a**) different values of the coefficient of friction ($m = 0.9$ (1 and 1′); $m = 0.7$ (2 and 2′); $m = 0.5$ (3 and 3′)) in the absence of holding elements and (**b**) different lengths of the holding elements ($H' = 0.9$ (1 and 1′); $H' = 0.7$ (2 and 2′); $H' = 0.5$ (3 and 3′)) for a fixed magnitude of the coefficient of friction ($m = 0.3$).

As discussed above, plastic deformation of the sample only occurs for the range of parameters where the double inequality (18) is fulfilled. This corresponds to the region between the two intercepts of the curves for $\omega_1'$ and $\omega_2'$ in Figure 3. The first intercept determines the beginning of the plastic twisting of the sample. As the sample moves through the container, the angular velocities of block 1 and block 2 rise. Initially, $\omega_2'$ grows faster, but, starting from $\xi = 0.5$, the growth of $\omega_1'$ becomes faster than that of $\omega_2'$. Eventually, at the second intercept of the curves $\omega_1'$ vs. $\xi$ and $\omega_2'$ vs. $\xi$, plastic twisting comes to a standstill.

Figure 3a illustrates the capacity of frictional forces with regard to the feasibility of HPTE without holding elements. For a fixed coefficient of friction ($m = 0.3$) the holding elements enable a longer part of a sample to undergo deformation by simple shear, as evident from a comparison between Figure 3a,b.

To verify the quality of the proposed mathematical model, we compare the distribution of the equivalent strain over the sample length following from Equation (5) with that calculated using the finite element software package QForm [32]. Two variants of the computations will be considered. To do this, consider two options for calculating. The first one corresponds to a real experiment conducted on copper samples [18]: $L = 35$ mm, $H = 24$ mm, $H_0 = 24$ mm, $R = 6$ mm, $m = 0.15$, $\omega = 1$ rpm and $\vartheta = 10$ mm/min. In the second variant, shorter holding elements ($H = 12$ mm) and a larger coefficient of friction ($m = 0.6$) are employed in the calculations. Both variants were simulated using finite element method (FEM) modelling. The upper and the bottom containers as well as the punch were defined as rigid bodies, whereas the workpiece was represented by 30,000 tetragonal deformable elements. Adaptive meshing was also used for the workpiece. The material behavior was taken to be isotropic using the von Mises perfect plasticity model. Friction between the billet and the container walls was modelled by the plastic friction law.

Perfect plasticity was used both in the analytical model and in the FEM simulations. This was done to provide a common ground for both modelling approaches and enable a comparison between the predictions of the two models. Furthermore, in [23] the authors looked into the influence of the choice of the material model on the deformation state of the workpiece and showed that the details of the constitutive law can be neglected in a first approximation. In this connection it should also be mentioned that experimental data for SPD exhibit a trend of the mechanical properties towards a steady state, thus justifying perfect plasticity as an adequate approach. One should recognise, however,

that in some cases (see, e.g., [33]) the rheology can have a strong influence on the deformation state of the workpiece leading to such effects as strain localisation. Investigations of such effects are enabled by simple rheological models, as suggested, for example, in [34,35]. An analysis of the role of rheology in the deformation behaviour of a workpiece under HPTE processing will be a subject of future research.

The equivalent strain distributions in the transversal cross-section through the middle of the sample for five consecutive points in time are presented in Figure 4.

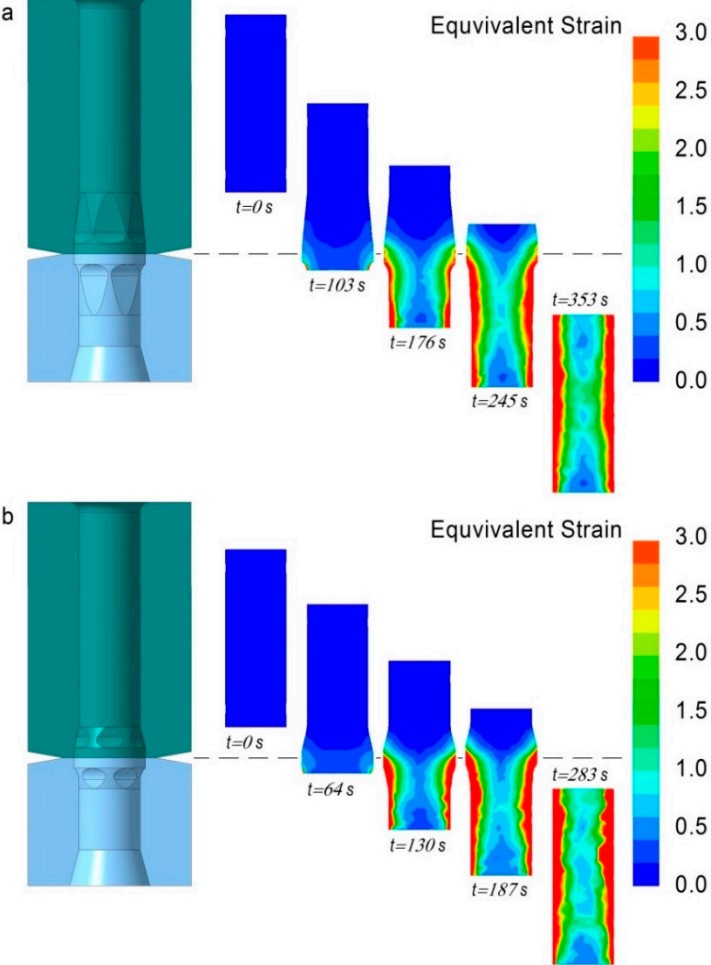

**Figure 4.** The equivalent strain distributions in the transversal cross-section through the middle of the sample calculated by (**a**) variant 1 and (**b**) variant 2 by means of QForm.

Figure 4 illustrates the accumulation of strain progressing with the movement of the sample through the holding elements. It clearly shows the non-uniformity of strain across the bulk of the sample. The dependence of the non-uniformity on the sample radius is associated with the increase of strain with the distance from the axis to the periphery of the sample, which is inherent in the torsion-based processes and is common to HPT and HPTE. Furthermore, near the surface the sample undergoes additional deformation through the cusps on the holding elements.

Non-uniformity of strain over the length of a sample is associated with a gradual increase of the counter-torque with the sample continually filling the holding elements, compounded with an insufficient capacity of the frictional forces at the beginning and the end of the process. The two variants considered differ in the contribution of these factors to the formation of the longitudinal non-uniformity. In the first variant (Figure 4a), the holding elements provide the torque necessary for plastic twisting. This is enabled by their sufficiently large length, which leads to the relative size of the end zone being small. In the second variant (Figure 4b), the gripping of the sample is established by a

concerted action of the frictional forces and the holding elements. As their length is relatively small, the strain non-uniformity along the sample caused by the holding element size is reduced. Given the sufficiently large capacity of the frictional forces, the uniform part of the sample is large in relation to its length.

The above statements are supported by the diagrams in Figure 5 which display equivalent strain distributions over the length of a sample at $r = 0.5R$.

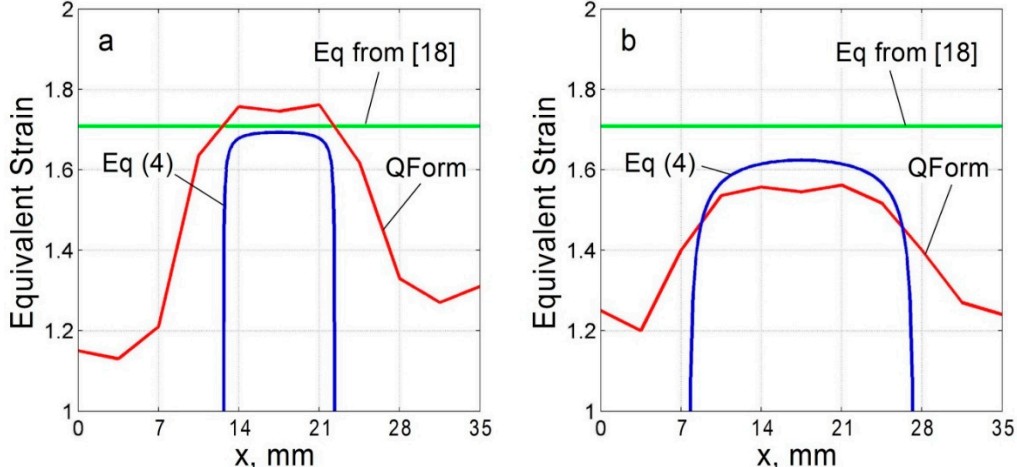

**Figure 5.** Dependence of the equivalent strain on the distance $x$ from the sample end surface at $r = 0.5R$: (**a**) variant 1 and (**b**) variant 2. The calculations done by using Equation (4) from the present work and the finite element simulations with QForm are seen to compare reasonably well. A constant level of equivalent strain calculated in [18] is also shown as a reference.

As seen in Figure 5, utilizing the friction force capacity in combination with the use of holding elements enables broadening of the uniformly deformed part of the sample. However, the magnitude of the equivalent strain in this portion of the sample is somewhat reduced. This is caused by slippage of the sample, which occurs in the case presented in Figure 5b. In the absence of slippage, the maximum values of the equivalent strain as predicted by the analytical model coincide with those obtained from the simple relation proposed in Reference [18] (cf. Figure 5a). The magnitude of the maximum equivalent strain in steady state computed using QForm (Micas Simulations Limited, Oxford, UK) (also shown in Figure 5) is somewhat larger. This is associated with some extra deformation in the sample parts contained within the holding elements.

Figure 5 also shows that the results obtained with the model outlined are in good agreement with the final element calculations, specifically those employing the QForm software. A precipitous drop of strain calculated using Equation (4) is caused by the model assumption that a holding element is to be fully filled with the deforming metal. In QForm, calculations are possible for partially filled holding elements, as well, which gives rise to descending parts of the graph.

Slippage of a sample relative to the containers leads to the emergence of a tangential near-surface flow of the metal within the holding elements. This gives rise to additional deformation with a shear plane aligned with the extrusion axis—an effect similar to the "cross flow" that occurs under Twist Extrusion [23]. A kinematically possible velocity field suggested in this study leads to a linear variation of strain along a sample radius. As shown in Reference [18], the strain distribution may deviate from a linear one. The current analytical modelling showed a good qualitative agreement with the results of FEM simulations with regard to the radial strain distribution, however. In addition, a good quantitative agreement was found for the regions at the periphery of a workpiece.

We consider the FEM simulation and the analytical model not as competing, but rather as being complementary to each other. The closed form analytical solution enables a large number of calculations for various combinations of the defining parameters at a low computational cost. This

makes it possible to carry out initial screening as a first step in the process optimisation. This should be followed by high-precision numerical calculations as a second step of the optimisation procedure. Thus, both modelling approaches would be brought to bear in this two-step optimisation process.

## 4. Conclusions

In this article, a simple mathematical model of the deformation behaviour of a rigid-plastic material in a severe plastic deformation process of high pressure torsion extrusion has been presented. The model provides a prediction of the worked part of a sample, even in cases when its slippage relative to the container walls occurs. It also predicts the magnitude of the equivalent strain of the material under such conditions. It is thus believed that the mathematical relations defining the window in the parameter space where an optimum in the HPTE performance is achievable will be useful for process design. The model proposed has also demonstrated that the use of holding elements and the resource of frictional forces to create a torque on a sample that makes it possible to maximise the length of its portion where nearly steady-state working of the material is achieved. We believe that due to its simplicity, generality and robustness, the mathematical model proposed can be successfully applied to calculating the mechanical behaviour of lightweight structures made from magnesium, aluminium, and titanium.

**Author Contributions:** R.K. and Y.B. conceived the idea presented in the paper. Y.B. developed the theory and R.K. performed the computations. Y.E. verified the analytical model. Y.I., B.B. and H.H. supervised the project. All authors discussed the results and contributed to the final manuscript.

**Funding:** This research was funded by German Research Foundation (DFG) grant number IV98/8-1 and by the Federal Ministry for Economic Affairs and Energy, based on the decision of a German Bundestag, IGF grant number 19838N.

**Acknowledgments:** The authors acknowledge funding support from the German Research Foundation (DFG) through Grant IV98/8-1 and partially by the Federal Ministry for Economic Affairs and Energy, based on a decision of the German Bundestag, IGF grant number 19838N.

**Conflicts of Interest:** The authors declare no conflict of interest.

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
