# Peer review of "A Mathematical Model of Deformation under High Pressure Torsion Extrusion"

_metals, doi:10.3390/met9030306_

Round 1

Reviewer 1 Report

The Authors present an analytical model for High Pressure Extrusion Torsion process.

The paper is very interesting and very well written, and should be considered for publication. 

In order to further improve it, I suggest to add some details about the description of the numerical FE model adopted to validate the proposed analytical solution. I think that these details should be added in order to make the FE model replicable.

In particular, what is the adopted constitutive model? Isotropic or kinematic hardening law or perfect plasticity? Friction model? Mesh? Which kind of analysis has been performed (explicit dynamic, static)?

I find two typos: 

Pag.7 row 192 share->shear

Pag. 9 row 235 he->the

Finally, a suggestion. The authors assume perfect plasticity to develop their model. In the literature, limit analysis solutions have been proposed also including linear hardening (see for example

A. Panteghini.   An analytical solution for the estimation of the drawing force in three dimensional plate drawing processes. Int J Mech Sci, 84C:147-157, 2014. DOI: 10.1016/j.ijmecsci.2014.04.012

A. Panteghini, F. Genna. An engineering analytical approach to the design of cold wire drawing processes for strain-hardening materialsInt J Mater Form, 3:279-289 )

 Do the Authors think that a similar approach could be used also in their model? In this way the material parameters could be better taken into account.

I suggest to accept this interesting paper after revision.

Author Response

Please find the attached file with answers.

Reviewer 2 Report

I did not find any, worthy of attention, novelty either in the theory of the description of the studied process, or in the procedure of numerical realization, or in the subject of the study. Therefore, I cannot recommend this article for publication in the journal.

Author Response

(The authors gave the same response as above.)

Reviewer 3 Report

High Pressure Torsion Extrusion (HPTE) was discussed by the authors and their merits were compared with other similar projects.

Plastic deformation is highlighted as a key processes to control the Multi-scale structures produced by SPD techniques. The authors correctly criticize similar processes for failing to prevent slippage at the sample tool interface contact surface which result in an ill-controlled process.

This is an interesting work and I suggest some minor modifications: the authors may want to revise their work by answering to the following questions

1-      It seems to me that HPTE suffers from the very same slippage issue. If this is so, is the HPTE’s analytical model the most significant part of this study?

2-      Deformation inside the test sample is heterogeneous. It changes between zero at the centreline to its maximum at the sample-tool interface. This only yields a non-unique kinematic solution. How one can evaluate this non-unique solution?

3-      The authors offer a mathematical model to optimize parameters of the HPTE process. Even an optimized design results in the non-uniform deformation, can this be an advantage for a given application? If so, could the authors provide some potential applications.

4-      Spiral extrusion processes of different cross sections have been proposed by several authors; twist extrusion, spiral forward extrusions with fixed and variable pitch and twist extrusion with oval cross section etc. How the proposed process and its associated model can be compared with those and what advantages they offer?

5-      The model is based on a pure shear deformation which neglects radial component of deformation (grooves at the interface). The assumption has to be stated in the paper; while this enables estimating the effective strain, the model is not detailed enough to evaluate the strength of material and structural design of the tools needed to perform HPTE. What are the advantages of the closed form solution compared to that of a numerical one?

Author Response

(The authors gave the same response as above.)

Round 2

Reviewer 1 Report

The manuscript has been improved. I have no further comments. I reccomend to accept it in present form 

Author Response

Thank you very much.

Reviewer 2 Report

I already wrote in the previous review that I did not find any, worthy of attention, novelty either in the theory of the description of the studied process, or in the procedure of numerical realization, or in the subject of the study. My opinion hasn't changed. I cannot recommend this article for publication in the journal and I do not see any point in further discussion of the submitted article with me.

Author Response

Thank you very much for your opinion.